# Pushing the Limits of Seagrass Remote Sensing in the Turbid Waters of Elkhorn Slough, California

**Heidi M. Dierssen** [1,2,*] , **Kelley J. Bostrom** [1], **Adam Chlus** [1], **Kamille Hammerstrom** [3], **David R. Thompson** [4] **and Zhongping Lee** [5]

1   Department of Marine Sciences, University of Connecticut, 1080 Shennecossett Rd, Groton, CT 06269, USA
2   Vlaams Instituut voor de Zee (VLIZ), 8400 Ostend, Belgium
3   Moss Landing Marine Laboratories, 8272 Moss Landing Road, Moss Landing, CA 95039, USA
4   Jet Propulsion Laboratory, California Institute of Technology, Pasadena, CA 91109, USA
5   Department of Environmental, Earth and Ocean Sciences, University of Massachusetts Boston, Morrissey Blvd., Boston, MA 02125, USA
*   Correspondence: heidi.dierssen@uconn.edu; Tel.: +1-860-405-9239

**Abstract:** Remote sensing imagery has been successfully used to map seagrass in clear waters, but here we evaluate the advantages and limitations of different remote sensing techniques to detect eelgrass in the tidal embayment of Elkhorn Slough, CA. Pseudo true-color imagery from Google Earth and broadband satellite imagery from Sentinel-2 allowed for detection of the various beds, but retrievals particularly in the deeper Vierra bed proved unreliable over time due to variable image quality and environmental conditions. Calibrated water-leaving reflectance spectrum from airborne hyperspectral imagery at 1-m resolution from the Portable Remote Imaging SpectroMeter (PRISM) revealed the extent of both shallow and deep eelgrass beds using the HOPE semi-analytical inversion model. The model was able to reveal subtle differences in spectral shape, even when remote sensing reflectance over the Vierra bed was not visibly distinguishable. Empirical methods exploiting the red edge of reflectance to differentiate submerged vegetation only retrieved the extent of shallow alongshore beds. The HOPE model also accurately retrieved the water column absorption properties, chlorophyll-a, and bathymetry but underestimated the particulate backscattering and suspended matter when benthic reflectance was represented as a horizontal eelgrass leaf. More accurate water column backscattering could be achieved by the use of a darker bottom spectrum representing an eelgrass canopy. These results illustrate how high quality atmospherically-corrected hyperspectral imagery can be used to map eelgrass beds, even in regions prone to sediment resuspension, and to quantify bathymetry and water quality.

**Keywords:** hyperspectral airborne imagery; eelgrass; optically shallow water; ocean color; prism; water quality; bathymetry

## 1. Introduction

Remote sensing techniques are being increasingly employed to monitor the distribution of submerged aquatic vegetation, such as seagrass meadows. These can vary from employing semi-empirical classifications from multichannel images from high-resolution satellites [1–9] to semi-analytical algorithms from airborne hyperspectral imagery [4,10–12]. Multichannel broadband sensors like Landsat and Ikonos are quite useful to delineate the presence or absence of benthic vegetation at low cost, but they lack spectral information that is necessary for determining the benthic composition without a priori knowledge or coincident field campaigns [8]. Benthos covered by seagrass, macroalgae and even coral appear similar with a multichannel broadband sensor [3,13–15]. Airborne

campaigns are generally costly requiring field personnel and clear weather conditions, but they offer the advantage of combining both high spectral and high-spatial-resolution imagery with the potential to identify benthic constituents, bottom depth, water column turbidity, and additional parameters such as seagrass leaf area index. This contribution is aimed at evaluating the utility of hyperspectral airborne imagery for eelgrass remote sensing in optically complex waters.

Elkhorn Slough, one of the largest remaining coastal marshes in California, U.S.A, is a eutrophic embayment with turbid water due to twice-daily tidal flow producing "tidal scour" [16] and increasing nutrient loads [17,18]. Beds of eelgrass occur in patches along the sides and main channel of the Slough occurring in shallow (<1 m) to deeper regions (3 m). Seagrasses are highly sensitive to light availability, declining with even small reductions in water clarity, and are used as indicator species of coastal health or "coastal canaries" [19–21]. Despite poor water clarity, however, Elkhorn Slough has recently been shown to have expanding eelgrass meadows linked to top-down control with the return of the sea otters [18]. Through a trophic cascade, sea otters (top predator) consume and limit crab populations (mesopredators) which allow the sea slug population (mesograzers) to increase and clear leaves of epiphytic algae (competitors) that would otherwise decimate the eelgrass. Hence, this region is experiencing a changing seagrass distribution that requires accurate methods to track changes over time.

Various airborne campaigns have been conducted over this region for development of remote sensing methods using high-spatial-resolution imagery [22,23]. In such turbid water, only light from the top few meters can be detected by a sensor depending on the tidal cycle and the amount of suspended particles in the water column [24]. Similar to studies from other regions [25], initial attempts to map eelgrass in Elkhorn Slough with hyperspectral imagery were best achieved with supervised or image-based classification due to challenges with low signal-to-noise ratios and issues with atmospheric correction [26]. However, recent sensor technology and algorithm advancements have greatly improved the quality of airborne imagery [27–29]. This has also allowed for better characterization of the atmospheric constituents including solar irradiance [30,31] and techniques for improved calibration of airborne sensors [32].

Advancing technology has also allowed for more sophisticated modeling of seagrass beds. Similar to terrestrial canopies, seagrass can be quantified with the metric Leaf Area Index (LAI), which is a measure of total leaf area per area of seafloor. Beyond presence or absence, this metric has been shown to be related to the net primary productivity of the canopy [3,33]. Radiative transfer models can now incorporate seagrass as a three-dimensional canopy in a moving fluid [34,35]. Sensitivity analyses based on leaf length, position, sediment coverage, and water depth revealed that there are limits to the density of the canopy that can be optically detected, and these limits are dependent on density and current flow that exposes more horizontal leaf area [4,11]. Modeling coupled with image analysis has also revealed that different genera of seagrass, such as the flat-leafed *Thalassia testudinum* and the cylindrical-leafed *Syringodium filiforme*, are too spectrally similar to be readily distinguished from hyperspectral remote sensing imagery [11].

Much of these advanced canopy models have proved useful in clear oligotrophic waters, but here we assess methods in a tidal embayment with shallow and dense eelgrass beds. While there is no present satellite sensor that can provide the required spectral, spatial, and temporal resolution to provide continuous monitoring capabilities for eelgrasses in this region [36,37], we evaluate various approaches and products that can be retrieved with airborne hyperspectral imagery with an eye towards development of future satellite missions and advancing drone technology. An advantage of using semi-analytical models is that they are able to simultaneously retrieve the water column optical properties and bathymetry, in addition to seafloor reflectance. To assess model performance, we compare methods for benthic mapping across different seagrass beds in Elkhorn Slough and also evaluate the accuracy of the retrieved water column optical properties and bathymetry compared to field data.

## 2. Materials and Methods

This section contains a description of the study site, collection and atmospheric correction of the hyperspectral imagery, water column, and benthic field measurements. Image processing and classification techniques are included as part of the results in Section 3.

### 2.1. Study Location

Elkhorn Slough is a 10 km long shallow tidally-forced embayment that extends nearly 5 km eastward from Monterey Bay before curving to the north [16] (Figure 1). Freshwater inputs to the slough are associated with seasonal episodic rainfall events and can be linked to extremely high inputs of nutrients due to extensive application of fertilizer within the watershed [38]. Water clarity in the main channel is influenced by the tidal cycle of the region which follows a mixed semi-diurnal tide with a tidal amplitude of ~2 m during our sampling period. On the flood tide, relatively clear water from Monterey Bay enters the slough. On the ebb tide, erosion of the dark gray bottom sediment reduces the water clarity [16,24]. The influence of tidal currents on water clarity is less pronounced on the shallow flats where eelgrass can provide a buffer to flow.

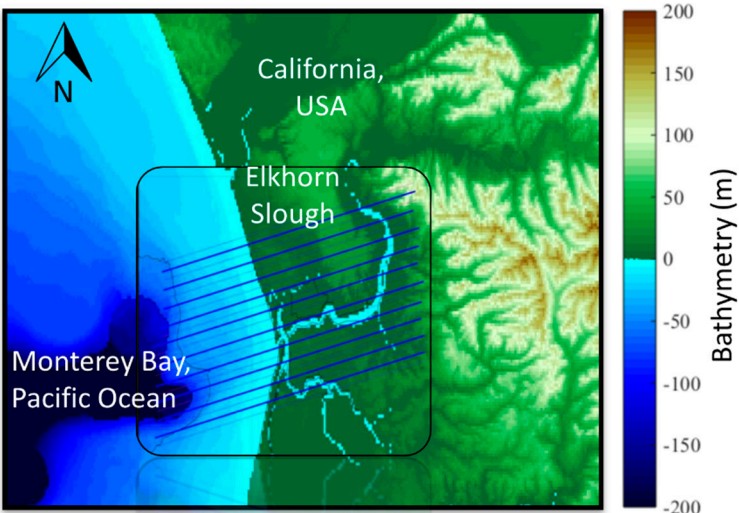

**Figure 1.** Elkhorn Slough, located along the California coastline at the head of Monterey Bay, is shown with gridded topography and the airborne tracks of PRISM imagery aligned in the plane of the sun to avoid cross-track illumination.

Elkhorn Slough contains a deep, swift-flowing central channel flanked by wider, shallow areas with aggregated sediment and slower tidal flow. This study focuses on an approximately 2 km² region of Elkhorn Slough, California, in the southwest near the mouth and includes Seal Bend, one of four major curves in the slough (Figure 2). Dense monospecific eelgrass meadows are known to occur along the shoals in the lower main channel around Seal Bend, Otter curve, Seal Alley, and two locations closer to the mouth of the Slough named Oyster Alley and Vierra, which experience higher current flows. Depths in this region were measured from sidescan sonar collected from the Seafloor Mapping Lab, California State University Monterey Bay [39] (Figure 3). At high slack tide, water depths at the Seal Bend, Seal Alley, Otter Curve and Oyster Alley were around 1.5–2 m. Water depth at the Vierra Bed ranged from 2 m nearshore to 4.5 m in the central channel. The upper limit of eelgrass extent is related to the tidal range and the danger of desiccation at the lowest tides [40]. The lower limit of eelgrass is related to water clarity and light limitation at depth. In this region, the slough extends 200 m across, and the main channel reaches depths of 10 m.

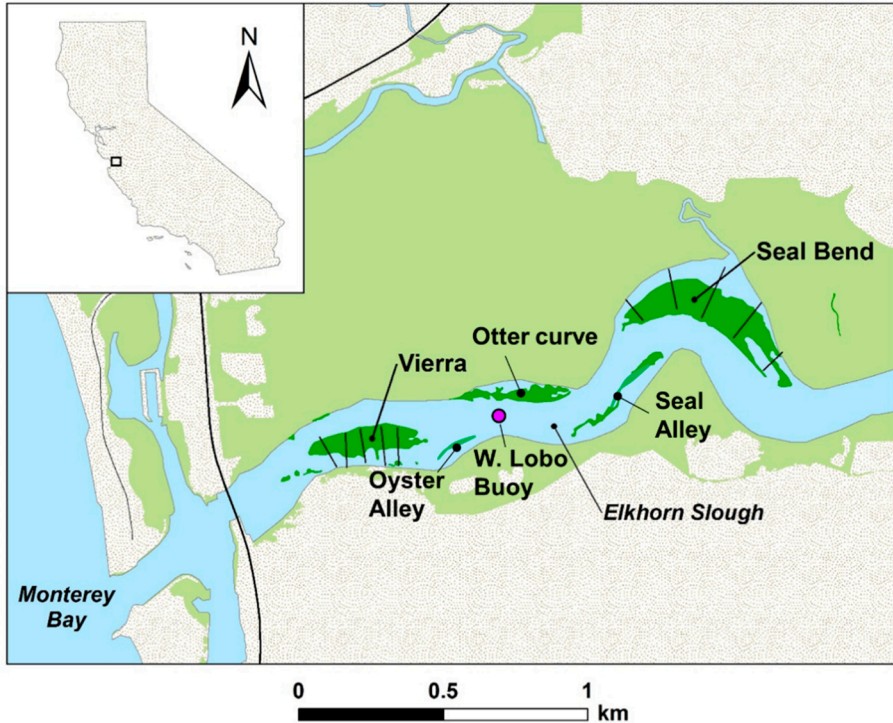

**Figure 2.** A highlighted portion of Elkhorn Slough with the identification of 5 different eelgrass beds considered in this study. Field data (PR019-PR022) was collected coincident with the PRISM overflight from a boat next to the West LOBO buoy, identified with a magenta circle. The routine monitoring transects for the eelgrass beds at Seal Bend and Vierra are shown as black lines adapted from [40].

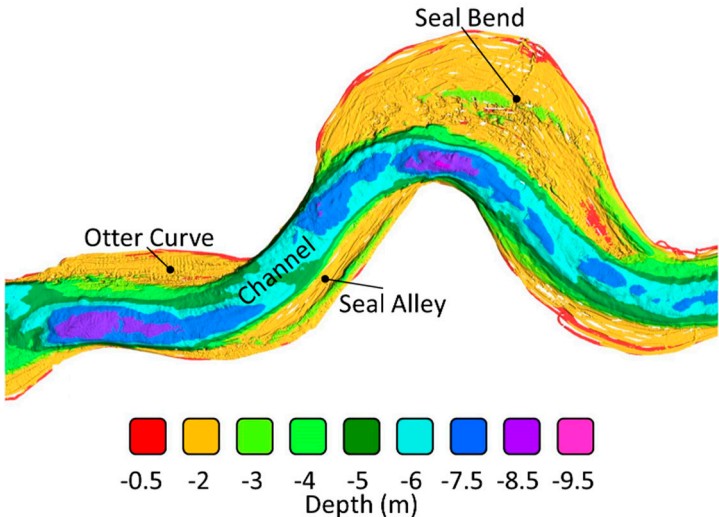

**Figure 3.** Bathymetry of Elkhorn Slough at the time of image collection corresponding to high slack tide. Eelgrass beds at Seal Bend, Otter Curve, and Oyster Alley occur in the shallower depths ~2 m. A deep channel runs through the center of Elkhorn Slough with higher current flow.

### 2.2. Field Data

The PRISM validation field campaign occurred during 18–27 July 2012. Using boats contracted from the small boat facilities at Moss Landing Marine Laboratories, surface measurement and dive teams conducted multiple surveys in the waters of the slough and in Monterey Bay. Field measurements coincident with the PRISM overflight on 24 July 2012 were obtained from a boat located in close proximity to the West LOBO Buoy (Figure 2). At this location, field spectra and profiled water column measurements were made in half hour intervals over a two-hour interval (stations PR019, PR020,

PR021, and PR022). Station PR022 was measured at 22:15 GMT (15:15 PDT) at the closest time period to the overflight, which passed over at 22:11 GMT. This corresponded to a high slack tide of 1.49 m above mean low low water (MLLW).

Apparent and inherent optical properties were measured concurrent with the overflight. In-situ inherent optical property profiles, salinity, and temperature were measured using a cage with a WET Labs ac-9, WET Labs ac-s, Seabird CTD, and WET Labs BBFL2 backscattering sensor (Philomath, OR USA). The ac-s was deployed to collect hyperspectral measurements of absorption and attenuation by particulate and dissolved material, $a_{pg}(\lambda)$ and $c_{pg}(\lambda)$ [m$^1$]. The ac-9 was outfitted with a 0.2 μm filter to measure the coincident dissolved component of absorption, $a_g(\lambda)$ and derive the particulate absorption $a_p$[m$^{-1}$] [41]. Corrections for the temperature and salinity dependence of pure water were applied to the absorption and attenuation measurements following [42]. The proportional correction method was employed to minimize scattering error in the measurements of absorption [43]. An estimate of the spectral absorption coefficients of phytoplankton ($a_{ph}$) and non-algal particles ($a_d$) was derived from particulate absorption ($a_p$) following the methods of Roesler et al. [44] and Schofield et al. [45]. The decomposition presumed a spectral slope of non-algal particulate absorption of 0.011, and the ratio of a$_{ph}$ at 412 nm to 440 nm was 0.9.

Particulate scattering in the backward direction, $b_{bp}$ (m$^{-1}$), was determined from measurements of the volume scattering function, β (m$^{-1}$·sr$^{-1}$), at 660 nm using a BBFL2 (WET Labs, Philomath, OR USA). Measurements of $β(660)$ were corrected for total absorption along the pathlength, and the volume scattering function of pure water was subtracted [46]. Particulate backscattering at 660 nm, $b_{bp}(660)$, was determined from $β_p(660)$ using a χ factor of 0.9 following Sullivan et al. [47].

With the exception of the benthic spectral library, field data were processed after image analysis and were used only for results comparison and not for calibration of the algorithm. The field data were further processed to validate water column parameters derived from the semi-analytical inversion algorithm HOPE (Hyperspectral Optimization Processing Exemplar) [48] with 5 variables (*P*, *G*, *X*, *B*, and H), namely: *P* is the absorption by phytoplankton at 440 nm ($a_{ph}(440)$); *G* is the absorption by colored dissolved organic matter (CDOM) at 440 nm ($a_g(440)$); *X* is the particulate backscattering at 550 nm ($b_{bp}(550)$); *B* is bottom reflectance at 550 nm; and *H* is bottom depth. The parameter *Y*, the spectral slope of particle backscattering, was not measured in this study. For comparison to model derivations, particulate backscattering at 550 nm was determined using the relationship $b_{bp}(550) = b_{bp}(660) (550/600)^{-Y}$, with *Y* presumed to be 0.5, in the range of that retrieved following Lee et al. [48].

Particle size distribution (PSD) measurements were taken with a LISST-100X, type B (Laser In-Situ Scattering and Transmissometry, Sequoia Scientific, Bellevue, WA USA) which uses laser diffraction to nominally determine the particle size distribution from 2 to 250 μm with a high sample rate (>1 Hz) and processed following [24]. The PSD slope represents the power law distribution of particles from small to large size classes fit using a least squares estimator of the log-transformed variables.

Above-surface radiance measurements were taken with the Field Spec Pro™ VNIR-NIR1 portable spectrometer system (Analytical Spectral Devices, Inc., Longmont, CO, USA). A sequence of measurements was made with an 8° foreoptic focused at a 40–45 degree angle sequentially on a gray plaque, sea surface, and sky following Dierssen et al. [3]. Residual reflected sky radiance was corrected using reflectance differences between 715 and 735 nm; an approach specifically developed for coastal waters [49]. Water samples were collected from each station for chlorophyll-a fluorescence, High Pressure Liquid Chromatography (HPLC), and Total Suspended Matter (TSM) analysis. Replicate samples for TSM were passed through pre-weighed 0.4 μm polycarbonate (Poretics) filters. After filtration, filters were rinsed with distilled water to remove salts and frozen for later gravimetric analysis. Filters for HPLC analysis were collected and stored following NASA protocols and analyzed by contract at Horn Point Laboratory.

Diver sampling was conducted on 23 July at slack low tide and on 25 and 26 July 2012 at slack high tide. Diver sampling included eelgrass distribution and abundance assessment by measuring shoot density at meter scale using transects placed underwater. Leaves were collected from the field

for morphometric and radiometric analyses in the laboratory. Leaf Area Index was calculated from the average shoot density and the average one-sided leaf area per shoot as determined by the morphometric measurements. Benthic reflectance, $R_b(\lambda)$, of hard substrate was assessed using the DiveSpec underwater Spectrometer provided by NightSea, LLC (Lexington, MA USA). The spectrometer was normalized to a spectralon plaque set on the substrate. Photographs were gathered of representative canopy cover from above and below water surface. Reflectance of individual eelgrass leaves were measured in the laboratory using the Field Spec Pro outfit with an integrating sphere. The mean of 5 repeat measurements from each leaf sample was presented here.

In addition, long-term seagrass monitoring data from Seal Bend and Vierra beds were obtained from the year prior (August 2011) and the year after the overflight (July 2013). Submerged aquatic vegetation monitoring was conducted following the National Estuarine Research Reserves protocol [50] for non-destructive annual sampling of percent cover, shoot density, and canopy height for each species present in each sampling plot along established transects. As shown in Figure 2, five transects were established at each site with multiple plots along each transect. Transects were oriented perpendicular to the edge of the bed, and endpoints at the deeper, channel-ward edge were determined by examining aerial photographs of the beds. In order to achieve good spatial coverage, transects were spaced approximately 100–150 m apart at Seal Bend and 50–75 m apart at Vierra [40]. Mean eelgrass percent cover values were calculated from the 2011 and 2013 to compare to the July 2012 imagery.

## 2.3. Google Earth and Sentinel 2 Imagery

Pseudo-true color imagery with Red, Green and Blue channels (RGB) was downloaded from the Google Earth archive for the years prior to and following the field campaign. The imagery contrast of each image was adjusted to highlight the eelgrass features. Comparisons were made to the known distribution of the 5 different named eelgrass beds visible in from field sampling and the airborne imagery. Visual interpretation was used to assess the presence (1), partial presence (0.5), or absence (0) of the five different beds from the images from 10/2011, 05/2012, 05/2013, and 08/2013. Sentinel 2A was launched after the field study, but here we evaluate the three earliest images of Elkhorn Slough from 18 September 2015, 11 November 2015, and 26 January 2016 for comparative purposes. Imagery was processed with ACOLITE (version 20190326.0, Brussels, Belgium) using the Dark Spectrum Fitting algorithm to compensate for atmospheric and surface effects [51]. Land areas were masked as having values in the 2202 nm band greater than a threshold of $R_{wl} = 0.0215$.

## 2.4. Hyperspectral Airborne Imagery

On a clear sky day of 24 July 2012, airborne imagery and coincident field spectral measurements of the waters within Elkhorn Slough were obtained as well as those extending out into the shelf of Monterey Bay. The PRISM sensor made 12 lines over Elkhorn Slough and the Shelf at an altitude of 1066 m (1 m/pixel resolution) (see Figure 1), 2 flight lines at 3200 m (3 m/pixel) extending 40 km into the bay, and a line over the M1 mooring at 3200 m (data not shown). The PRISM sensor provided 400 GB of raw imagery that was processed by NASA JPL for orthorectification and spectroradiometric calibration. This contribution focuses on the imagery obtained over the entrance to Elkhorn Slough on 24 July 2012 obtained at GMT 22:10:57 labeled 'prm20120724t221057'. During the overflight, the solar zenith angle was 30.8°, and the solar azimuth angle was 244.2° from due north (0°), a similar angle to the one the plane flew to obtain imagery up the Slough channel and minimize cross track sun glint.

Atmospheric correction is the process by which the photons scattered within the atmosphere are removed from the image spectra as well as any skylight that has glinted off the sea surface. The retrieval first estimates the Lambertian-equivalent water-leaving reflectance directly above the water surface, $R_{wl}$. The quantity is related to the common Remote Sensing Reflectance $R_{rs}$ by $R_{wl} = \pi\, R_{rs}$. Our $R_{rs}$ retrieval uses a modified version of the ATmospheric REMoval (ATREM, Washington, DC, USA) approach by Gao et al. [30,52]. It accounts for solar and atmospheric properties including irradiance, atmospheric absorption, and atmospheric scattering.

Atmospheric correction begins with a spectral image of calibrated radiances at each known wavelength channel and normalizes these radiances for variable solar illumination. Slight mischaracterizations of the solar irradiance and the instrument spectral response can cause noise in the narrow spectral bands of 3 nm in the PRISM sensor. Therefore, we applied a procedure that adjusts the solar irradiance spectrum to fit a spectrally smooth reference [31]. The revised solar irradiance F and solar incidence angle ψ relate measured radiances to at-sensor reflectance, $\rho$, via:

$$\rho = (\pi \text{ L})/(\text{F } \cos(\psi)) \tag{1}$$

Neglecting coupling between absorption and scattering, $\rho$ is approximately related to the apparent water-leaving reflectance, $R_{wl}$, by:

$$R_{wl} = (\rho/T_g - R_a)/(T_d \, T_u + S_a(\rho/T_g - R_a)) \tag{2}$$

where $T_g$ is the gaseous transmission of the atmosphere, $T_u$ and $T_d$ are upward and downward transmission due to scattering, $S_a$ is the spherical sky albedo, and $R_a$ is the path reflectance due to scattering. Scattering terms incorporate molecular (Rayleigh) scattering and aerosol scattering. We calculate these coefficients in advance using the 6s code [53] and a model atmosphere. In general, water absorbs nearly all irradiant energy at 1000 nm, so any finite observed radiance at these wavelengths is often due to a spectrally-constant glint reflectance. We subtract the value at this wavelength to yield a glint-corrected reflectance and divide by $\pi$ to retrieve remote sensing reflectance, $R_{rs}(\lambda)$.

### 2.5. Benthic Retrieval Algorithms

#### 2.5.1. Red Edge Retrieval

Many algorithms for using remote sensing to detect submerged aquatic vegetation (SAV) often rely on decision trees with different ratios and defined thresholds to distinguish different benthic types [33,54]. These approaches often use different wavelengths in a normalized difference index (NDI) to screen optically deep water from vegetated pixels:

$$NDI = \frac{R_{rs}(\lambda_1) - R_{rs}(\lambda_2)}{R_{rs}(\lambda_1) + R_{rs}(\lambda_2)}. \tag{3}$$

Here, we implemented an algorithm to classify vegetated marine targets using the red edge of submerged and floating vegetation where $\lambda_1 = 700$ nm and $\lambda_2 = 670$ such that values greater than 0 were considered vegetated and less than 0 predominantly sand or optically deep [33].

#### 2.5.2. Semi-Analytical Inversion Method

Dekker et al. (2011) presents a summary of various approaches to estimate bathymetry and benthic composition using semi-analytical methods [10]. Here, we implemented the HOPE model on the atmospherically corrected imagery [48]. First, the remotely observed $R_{rs}$ was extrapolated across the air-water interface to below the water surface, written $r_{rs}$:

$$r_{rs} = (R_{rs})/(1.562 \, (R_{rs}) + 0.518). \tag{4}$$

As mentioned above, the HOPE scheme inverts this spectrum into predominantly five different parameters: X, P, G, B, and H, which modulate the total backscattering ($b_b$) and absorption ($a$) coefficient spectra. We model $b_b$ by decomposing it into a sum of backscatter due to water, $b_{bw}$, and particulates, $b_{bp}$. We calculate $b_{bw}$ from Morel [46], and the particulate term using the formulation of Lee et al. [48]. The parameter $X$ represents particle backscattering at 550 nm ($b_{bp}(550)$). The spectral slope of backscattering, $Y$, was held to the [0, 2.5] interval and a retrieved free parameter X was estimated such that:

$$b_{bp} = X \, (400/\lambda)^{-Y} \tag{5}$$

$$Y = -3.44 \, (1 - 3.17 e^{(-2.01\,\chi)}) \tag{6}$$

$$\chi = r_{rs}(440)/r_{rs}(490). \tag{7}$$

We calculate total absorption coefficient as the sum of absorption coefficients by water, $a_w$, from Pope and Fry [55], phytoplankton, $a_{ph}$, and dissolved and detrital matter $a_{dg}$:

$$a = a_w + a_{ph} + a_g \tag{8}$$

$$a_{dg} = G \, e^{\, -0.015 \, (\lambda - 440)} \tag{9}$$

$$a_{ph} = \phi_1 \, P + \phi_2 \, P \, ln \, P \tag{10}$$

Here G and P are free parameters related to the concentration of gelbstoff/detrital absorption and phytoplankton, respectively. The spectrally dependent coefficients $\phi_1$ and $\phi_2$ are provided in [48]. Following that work, the abundance of chlorophyll-*a* ($Chl_a$) in units of mg m$^{-3}$ was estimated as approximately:

$$Chl_a = e^{ln \, (P/0.06)/0.65} \tag{11}$$

The retrieval ultimately aims to determine benthic reflectance $R_b$. It is important to constrain the range of possible $R_b$ spectra to avoid the intrinsic ambiguity between bottom reflectance and water properties following Thompson et al. [56]. Here, we refine the parameterization of *B* as a pure endmember of a selected bottom type in the HOPE model and treat $R_b$ as a linear, or areal, mixture based on a library of known endmembers. Such linear unmixing has also been incorporated by other researchers using the HOPE model (e.g., [57–59]). These endmembers can be measured in situ or drawn from a suitable spectral library. We represent them as columns of a matrix **U**, with a length-n column vector **v** of nonnegative mixing coefficients. In this demonstration, the endmembers are three spectra of eelgrass leaves and five examples of sand and sediment from in-situ measurements. The bottom reflectances are combinations of endmembers in proportion to mixing fractions **v**:

$$R_b = \mathbf{U}\mathbf{v} \tag{12}$$

In summary, the retrieval reproduces the spectral observation using the following free parameters: The backscatter *X*, the gelbstoff/detritus absorption *G*, the phytoplankton absorption *P*, the bottom depth *H*, and the vector **v** of nonnegative mixture fractions. We use the Levenberg–Maquardt algorithm to optimize the Sum Squared Error (SSE) fit between the model spectrum $R_{rs}{'}$ and the PRISM observation $R_{rs}$. In practice, this minimization can fall into local minima. To improve stability, we introduce a regularization term for each free parameter $\theta_i$ including nonnegative optical parameters, mixture fractions, and depth. Each parameter is associated with an expected value $\mu_i$ and standard deviation $\sigma_i$. The free parameters are constrained conservatively within wide standard deviations to avoid biasing the results, and based on the ranges observed in the literature (e.g., Lee et al. 1998 [48]). Similarly, $\sigma_\lambda$ represents the standard deviation of measurement noise at $R_{rs}(\lambda)$. We calculated instrument noise based on the average variability in differences in across neighboring channels of featureless Near-Infrared regions of the Rrs spectrum. The resulting signal to noise ratio of 1000 was consistent with those reported for the PRISM instrument by Mouroulis et al. [29] after accounting for downtrack binning. We treated the noise matrix as diagonal and uncorrelated, ignoring the covariance structure for simplicity and because the diagonal captures the main noise-related effects as measured in the laboratory. The resulting SSE function is:

$$f\,(\theta) = \Sigma_\lambda \, [(R_{rs}{'}(\lambda, \theta) - R_{rs}(\lambda))/\sigma_\lambda]^2 + \Sigma_i \, [(\theta_i - \mu_i)/\sigma_i]^2 \tag{13}$$

Since the parameters can vary over the $[0, \infty]$ interval, we treat the optimization as an unconstrained minimization of the log parameters. For this analysis, the spectral shape and magnitude of the retrieved benthic reflectance was used to characterize the bottom as either wholly eelgrass or sediment, as highlighted in the results.

## 3. Results and Discussion

Remote sensing of optically shallow water is dependent on the constituents on the seafloor, the clarity of the water column, and the bathymetry. The results are presented and discussed for the field characterization of the eelgrass beds in Elkhorn Slough and the overlying water column to provide a context for parameterizing remote sensing algorithms for these estuarine waters. Then, the retrievals of eelgrass with simple RGB and hyperspectral imagery are compared using different algorithms and assumptions. The final analysis is conducted in the vicinity of the Vierra eelgrass bed where field measurements of water column properties were taken nearby coincident with the hyperspectral aircraft imagery.

### 3.1. Field Characterization of the Benthos

The seagrass *Zostera marina* L., commonly referred to as "eelgrass," is the dominant species found in dense monospecific meadows in the Elkhorn Slough watershed [40]. Eelgrass quantified in Seal Bend had on average 5–6 leaves per shoot with lengths averaging 56 ± 22 cm and leaf widths averaging 0.9 ± 0.1 cm. Each shoot had on average 0.025 ± 0.015 m² of one-sided leaf area. Shoot densities varied from 150–275 shoots per m² with resulting leaf area indices (LAI) averaging 5.63 ± 1.14 (Figure 4). These results are similar to measurements taken in Otter Curve in 2006 where LAI in the meadows ranged from 3.0–8.3 [22] and to long-term monitoring efforts that characterized similar ranges in shoot density averaging 156 ± 84 in Seal Bend and Vierra beds for 2011–2013. The LAI values measured in Elkhorn Slough are high compared to some of the tropical seagrass meadows in Florida and the Bahamas [3,4,33,60], and the beds are considered to be dense seagrass canopies where the underlying sediment is not generally visible.

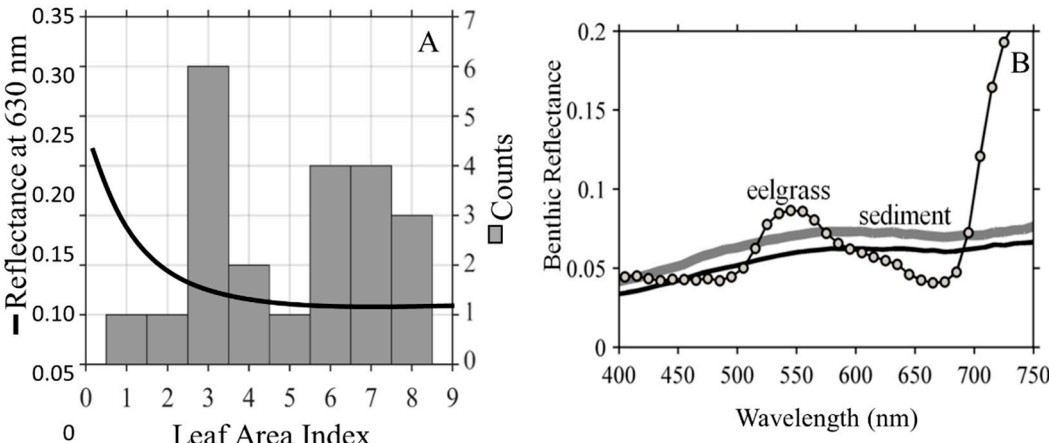

**Figure 4.** (**A**) A histogram of the eelgrass Leaf Area Index (LAI) measured along the various transects in Otter Curve and Seal Bend (left axis) with LAI values largely greater than 3 in this region. The solid black line represents the modeled relationship between LAI and Reflectance at 630 nm (right axis) from Hedley et al. (2017) revealing that the signal saturates for dense eelgrass canopies with LAI's greater than ~3 [11]. (**B**) Benthic reflectance of sediment measured by divers both within and outside the eelgrass beds (solid lines) compared to eelgrass leaf reflectance measured in the laboratory (dotted line).

Recent modeling work has shown that the relationship between top of the canopy reflectance of light and LAI decreases exponentially with increasing LAI [4,11]. In other words, the color at the top of the canopy darkens rapidly with initially increases in LAI, but the canopy color "saturates" at

some density of LAI (Figure 4A). For canopies with LAI greater than about 3, the canopy reflectance is constant and remote sensing methods cannot be accurately used to quantify LAI without a high level of uncertainty. Given the length of the eelgrass leaves and high LAI, however, we did not attempt to quantify LAI from the imagery and focused only on detecting the presence or absence of dense eelgrass meadows and the extent of the beds in the turbid waters of Elkhorn Slough.

Sampling at the Seal Bend and Vierra bed revealed no significant differences in percent cover or shoot density between 2011 and 2013 for stations with eelgrass and including stations with only sediment. Some of the station-specific difference was observed from year to year due the challenges in repeat sampling of the same transects over time with divers subject to current and the inaccuracies of GPS positioning. However, comparison between all 112 stations suggests that no significant differences were observed in eelgrass extent or density during this time period.

### 3.2. Optical Properties of Seafloor and Water Column

The remote sensing signal obtained over optically shallow water is influenced by the reflectivity of the seafloor and the amount and optical properties of the intervening water column. The sediment in this part of the Slough was dark gray in color ranging from 3% to 7% reflectance across the visible spectrum (Figure 4B). Sediment spectra increased monotonically from 400 to 700 nm and showed only a slight dip at the chlorophyll-*a* absorption band at ~660 nm indicating low amounts of associated algae or biofilm. Sediment within the bed was 11% darker than sediment outside the bed likely due to enhanced organic detritus and other absorbing matter trapped within the eelgrass meadow. Reflectance from eelgrass leaves was similar to leaf reflectance spectra from turtle grass [3] and other seagrass species from the literature [11,61], peaking in the green wavelengths (550 nm) and containing a sharp increase in reflectance in the far red and infrared portion of the spectrum, the "red edge of reflectance" associated with all pigmented vegetation [62,63]. This near infrared signal is highly absorbed by water molecules and only emerges from the water column when the benthos is very shallow.

The water column measurements of the spectral absorbance due to particles and dissolved matter reveal the relatively high turbidity of the estuary (Figure 5). Overall, absorption was low compared to total attenuation which indicates a significant amount of light scattering (Figure 5A). Absorption by dissolved and particulate matters was high with $a_{dg}(440)$ equal to $0.37 \pm 0.09$ m$^{-1}$ and phytoplankton absorption at 440 nm of $0.19 \pm 0.041$ m$^{-1}$. Absorption by non-algal particles and CDOM was two times higher than for phytoplankton (Figure 5B). Chlorophyll-*a* concentrations from HPLC quantified during the overflight was 4.75 mg·m$^{-3}$, consistent with the fairly high phytoplankton absorption. Chlorophyll-*a* concentrations estimated from $a_{ph}(440)$ using the optical relationship of Equation (11) totaled 5.9 mg m$^{-3}$ with the uncertainty ranging from 4.0–7.9 mg·m$^{-3}$.

Profiling data from the LISST confirmed that the turbidity within the Slough was due to high amounts of light scattering from particulates rather than absorption by phytoplankton or colored dissolved organic matter. The values of $c_{pg}(660)$ greater than 2 m$^{-1}$ (Figure 6A) indicate the presence of considerable particulates and are higher than attenuation in turbid sediment whitings in the Bahamas [64] and in seagrass beds of Florida Bay [60]. Similarly, the mean and one standard deviation of particulate backscattering $b_{bp}(660)$ was $0.059 \pm 0.009$ m$^{-1}$, respectively. These values are high compared to average coastal water and yield a backscattering ratio of approximately 0.03 which is indicative of minerogenic sediment particles [65].

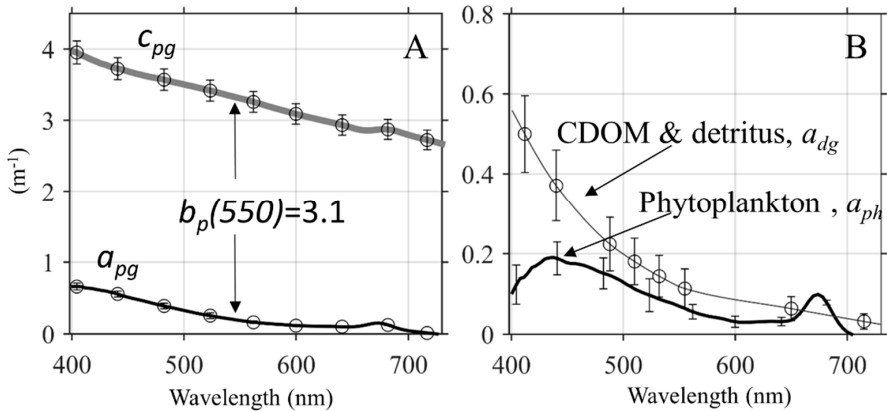

**Figure 5.** Field measurements near the West LOBO Buoy (Figure 1) made coincident with the overflights of (**A**) spectral attenuation ($c_{pg}$), absorption ($a_{pg}$) and particulate scattering ($b_p$). (**B**) Absorption measurements were partitioned into that by phytoplankton ($a_{ph}$) and colored dissolved and detrital matter ($a_{dg}$) using coincident measurements of $a_g$. The mean over approximately 1 h is shown with one standard deviation as error bars.

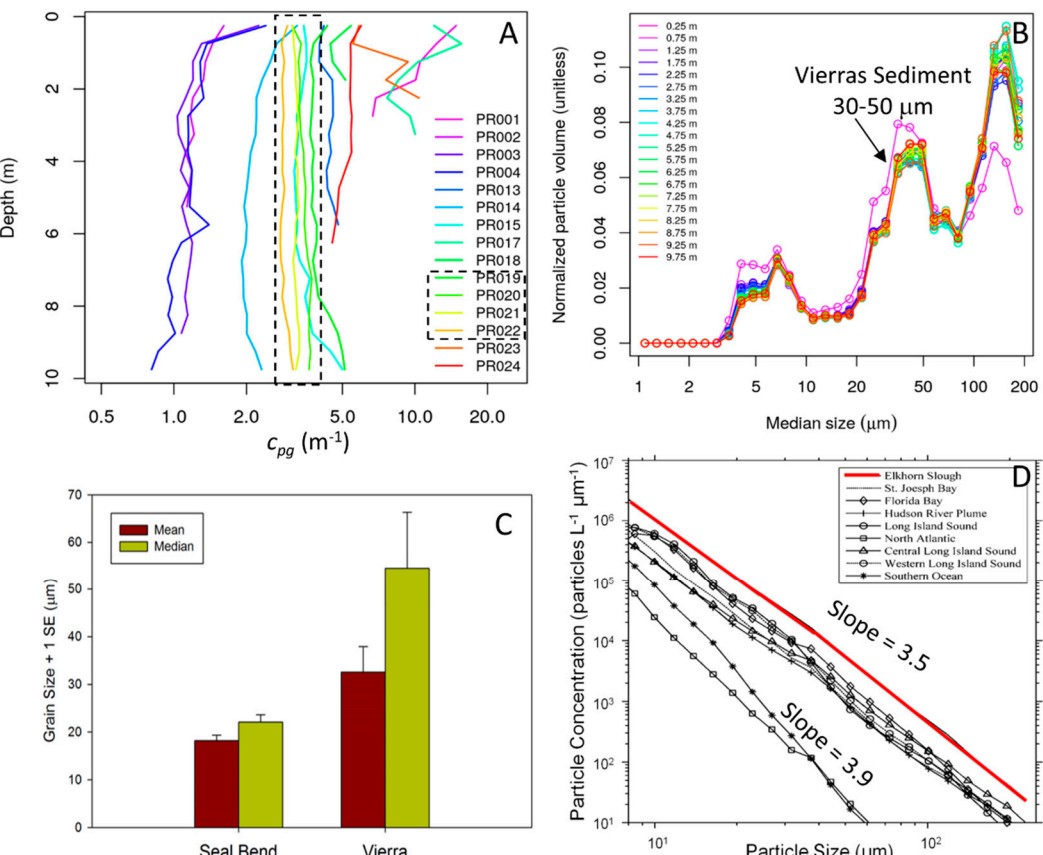

**Figure 6.** The optical properties in Elkhorn Slough are dominated by highly scattering particulate matter. (**A**) Measurements of light attenuation by particles and colored dissolved matter (cpg) from the LISST-100X show that the water column was well mixed during the stations coincident with the overflights (black box). (**B**) The particulate volumetric size distribution from the stations outlined in panel A reveals three predominant particle populations from 5–10 µm, 30–50 µm, and 120–150 µm. (**C**) The mean sediment particle sizes measured within two eelgrass beds reveals larger particles at the Vierra bed consistent with size of suspended particles measured with the LISST data from Panel B. Adapted from [45]. (**D**) A global comparison of particle size distributions shows the high amount of particulates and flatter Jungian slope in Elkhorn Slough compared to other sites. Adapted from [24].

During the overflights, the particle load in the main channel was well mixed down to 10 m and did not vary considerably over the time course of the overflights (Figure 6A). The prevalent particles in the water column were consistent with sediment grain sizes of the sediment near the Vierra bed with a mean of ~30 μm and a median of 50 μm [40] (Figure 6B,C). This is consistent with the presence of high amounts of sediment scoured from the seabed and resuspended into the water column. These data illustrate that Elkhorn Slough is a very turbid embayment with high concentrations of scattering suspended particles and light penetration of predominantly green light to only the top few meters in depth. Compared to other regions, the measured particle size distribution for Elkhorn Slough revealed higher loads of suspended particles compared to other sampled estuaries [24] (Figure 6D), and the power-law or Jungian slopes of the particle size distributions were flatter (3.5) than for the open ocean (3.9).

### 3.3. Google Earth Imagery

The ability to map the benthos from remote sensing reflectance is due to changes in the appearance of water color. Aerial photography measures the total upward radiance reflected from the sea surface. The detected radiance is not atmospherically corrected and is therefore influenced by the spectral intensity and magnitude of sunlight hitting the sea surface, as well as the reflections of both the direct solar beam (glint) and the diffuse skylight off of the sea surface. All of these factors influence the ability of numeric algorithms and human photo-interpretation to correctly identify the presence of eelgrass beds. Such reflections can be challenging to remove and can dominate the signal over dark water surfaces. The color observed from each photograph can be quite different depending on the time of day and the direction of the sensor angle and flight path in relationship to the sun. In general, seagrass meadows absorb considerable amounts of light for photosynthesis and appear as relative dark patches in imagery.

In order to illustrate the ability to use Google Earth to conduct time series analyses, RGB images were compared for Elkhorn Slough over a 2-year time period from 2011–2013 when the sampled eelgrass density was similar. The image quality dramatically influenced the ability to observe different eelgrass meadows, and each image had variable amounts of eelgrass visible (Figure 7). For example, the Vierra bed was not visible in 10/2011 and 05/2012. The partial darkening in the 05/2012 image over the Vierra bed resulted from a division across the whole Slough separating apparent light and dark regions. Approximately half of the Vierra bed was observable in 05/2013, and the full bed was apparent three months later in 08/2013. Table 1 reveals the apparent change in the five different eelgrass beds over time determined visually from the imagery. Even though field data suggests the Seal Bed and Vierra beds were similar over time, using broad-channel RGB imagery that has not been normalized for differing solar illumination and surface reflections made assessments of the beds highly variable over time.

**Table 1.** Retrieval of different seagrass beds in Elkhorn Slough from Google Earth time series.

| Region | Google Earth Image Date | | | | PRISM |
|---|---|---|---|---|---|
| | 10/2011 | 05/2012 | 05/2013 | 08/2013 | 07/2012 |
| Vierra | 0 [1] | 0.5 | 0.5 | 1 | 1 |
| Oyster Alley | 0 | 0 | 0.5 | 1 | 1 |
| Otter Curve | 1 | 1 | 1 | 1 | 1 |
| Seal Alley | 0 | 0.5 | 1 | 0.5 | 1 |
| Seal Bend | 1 | 1 | 1 | 0.5 | 1 |

[1] Key: 0 = Bed not visible, 0.5 = Bed partially visible, 1 = Bed fully visible.

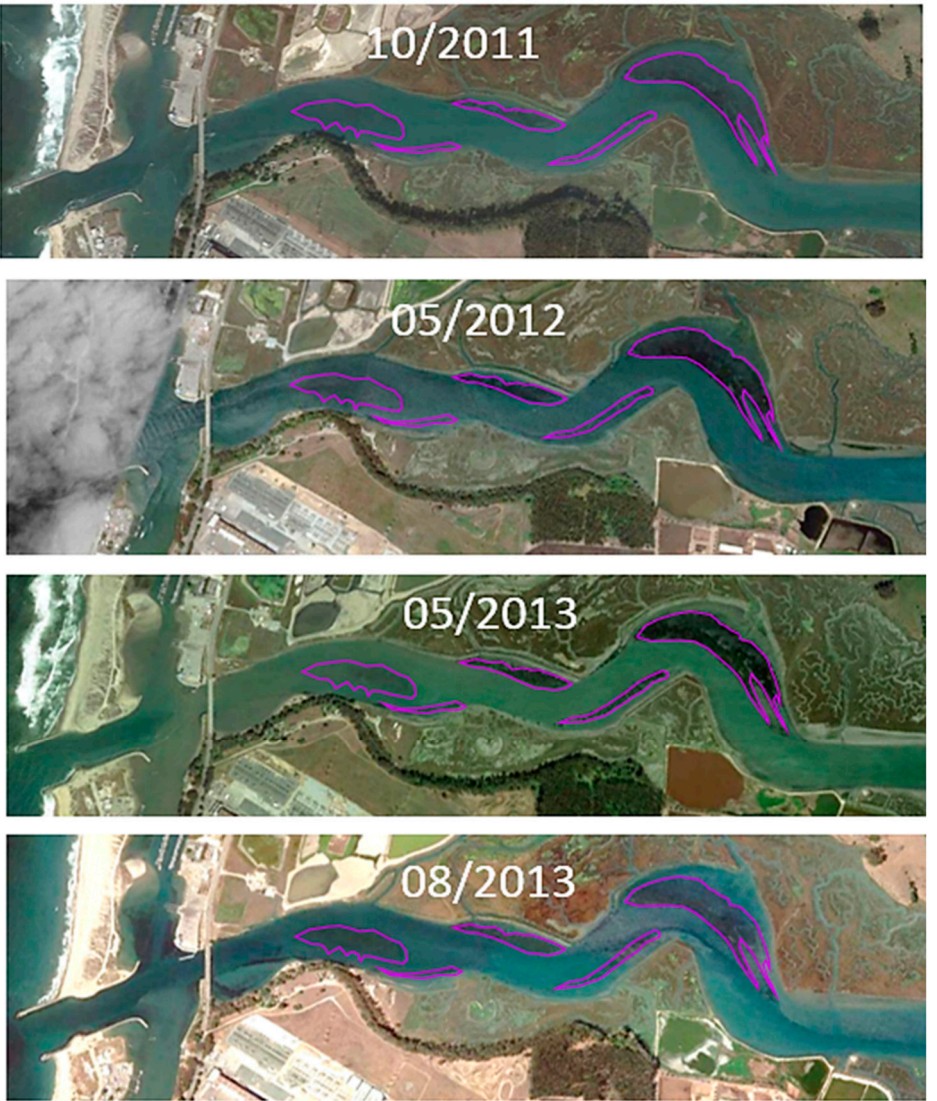

**Figure 7.** Comparison of available images from the Google Earth archive of Elkhorn Slough from October 2011 through August 2013. The magenta outlines provide the shapes of the five eelgrass beds revealed from the 2012 PRISM imagery.

### 3.4. Satellite and Aircraft Spectrometry

Different from aerial photography, ocean color remote sensing relies on properly calibrated sensors and imagery that have been corrected for atmospheric and sea surface effects. As described in the methods, the radiance measured at sensor is normalized to an estimate of the solar irradiance on the sea surface to derive a relative reflectance at each wavelength. Secondly, the imagery is "atmospherically corrected" to account for atmospheric absorption and scattering properties and direct and diffuse reflections of photons off of the sea surface (i.e., photons that are not impacted by the water properties). The resulting measurement is called Remote Sensing Reflectance ($R_{rs}$) and represents the water-leaving radiance normalized to the incident downwelling irradiance. This measure, also called water-leaving reflectance, quantifies the color without the air-water interface and should include only photons that have penetrated into the water column. For inversion modeling, this parameter is related to the reflectance beneath the sea surface (rrs) via Equation (4).

The launch of the multispectral Sentinel 2 satellite in 2015 with high spatial resolution was also demonstrated to be useful for mapping shallow water habitats. The number of bands is not sufficient to characterize both the benthos and water column using an algorithm like HOPE. However,

the imagery is calibrated and atmospherically corrected to allow for consistent, robust analyses of aquatic ecosystems. Specifically, the imagery is useful at delineating the extent of benthic features, like seagrass, if the regions of interest are well-characterized in advance. Here, images from approximately two years after this study revealed some portion of all five of the eelgrass beds indicating that the sensor has capabilities to monitor eelgrass in this turbid estuary when conditions are favorable. Evaluations of the Sentinel-2 imagery over time, however, show that the deeper Vierra bed was only intermittently visible likely due to tidal stage and water clarity. However, the lack of spectral information makes it challenging to differentiate the type of benthic flora (right panels in Figure 8) besides a relative darkening of the spectrum over the eelgrass beds. Simple wavelength thresholds, band ratios, or object based image analysis techniques can be used to differentiate the general location of the shallow beds.

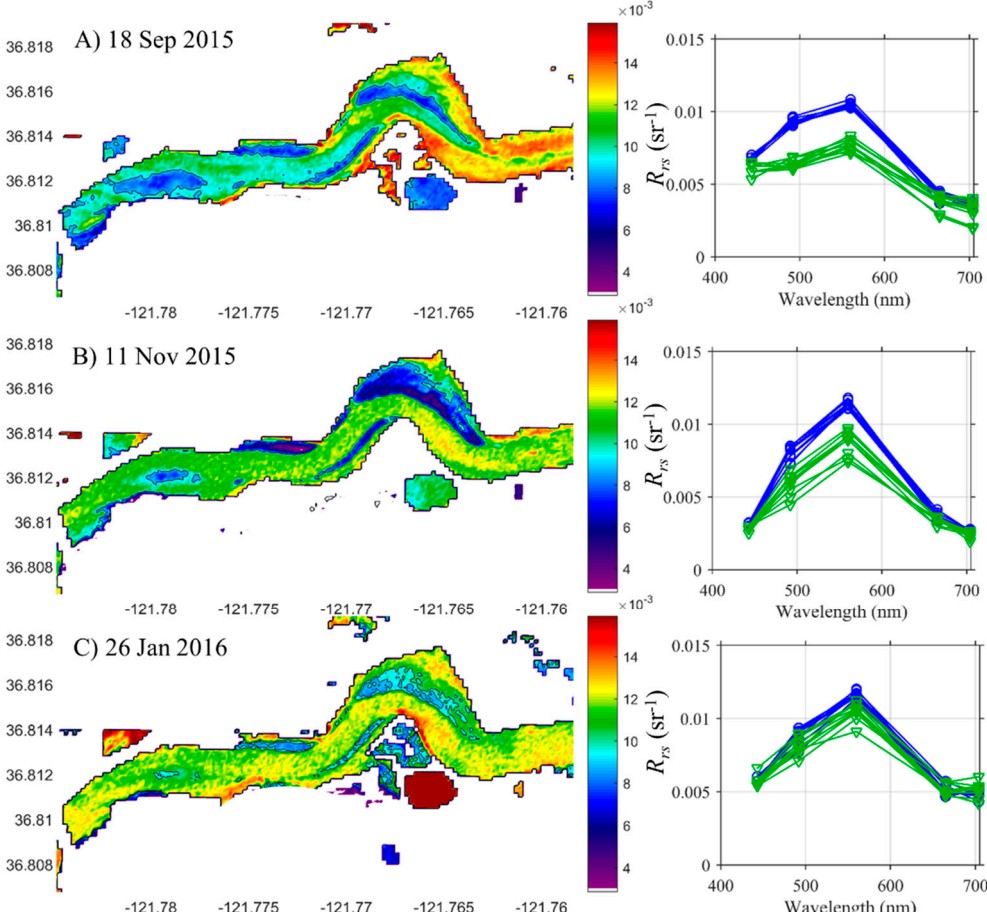

**Figure 8.** Sentinel-2 images obtained in 2015 and 2016 over Elkhorn Slough reveal darker patches related to the five eelgrass beds. Retrieved remote sensing reflectance at 560 nm can be used to quantify the extent of eelgrass beds based on low reflectance (<0.009) with black contour line provided. For panel C, an additional contour line at 0.01 was added. The right panels show the remote sensing reflectance from over the Vierra seagrass bed (green) compared to sediment seafloor near the W. Lobo buoy (blue) for each image.

The atmospherically-corrected PRISM imagery covers the entire visible spectrum at ~3 nm resolution allowing for more detailed analyses of spectral shape, in addition to spectral magnitude. For example, the spectra over the various eelgrass beds have a broader peak and are lower in magnitude than $R_{rs}$ measured over sediment (Figure 9A,B). The height of the spectrum also depends on the depth of the water column and hence the Seal Bend spectra are shallower with higher $R_{rs}$ than the deeper Vierra beds with lower $R_{rs}$. Normalized to the value at 555 nm, the spectral shape was lower in magnitude and exhibited a broader peak compared to spectra obtained over sediment or deep water

(Figure 9C). These spectra were also compared to an $R_{rs}$ spectrum obtained over sediment near the Vierra bed in the main channel nearly coincident with the overflight. As highlighted in Figure 9B, this independent field measurement was similar in both magnitude and shape to the $R_{rs}$ derived from the PRISM imagery, which provided confidence in the sensor calibration, data processing, and atmospheric correction of the PRISM imagery. An additional field match-up from a later tidal stage is found in Dierssen (2013) [23].

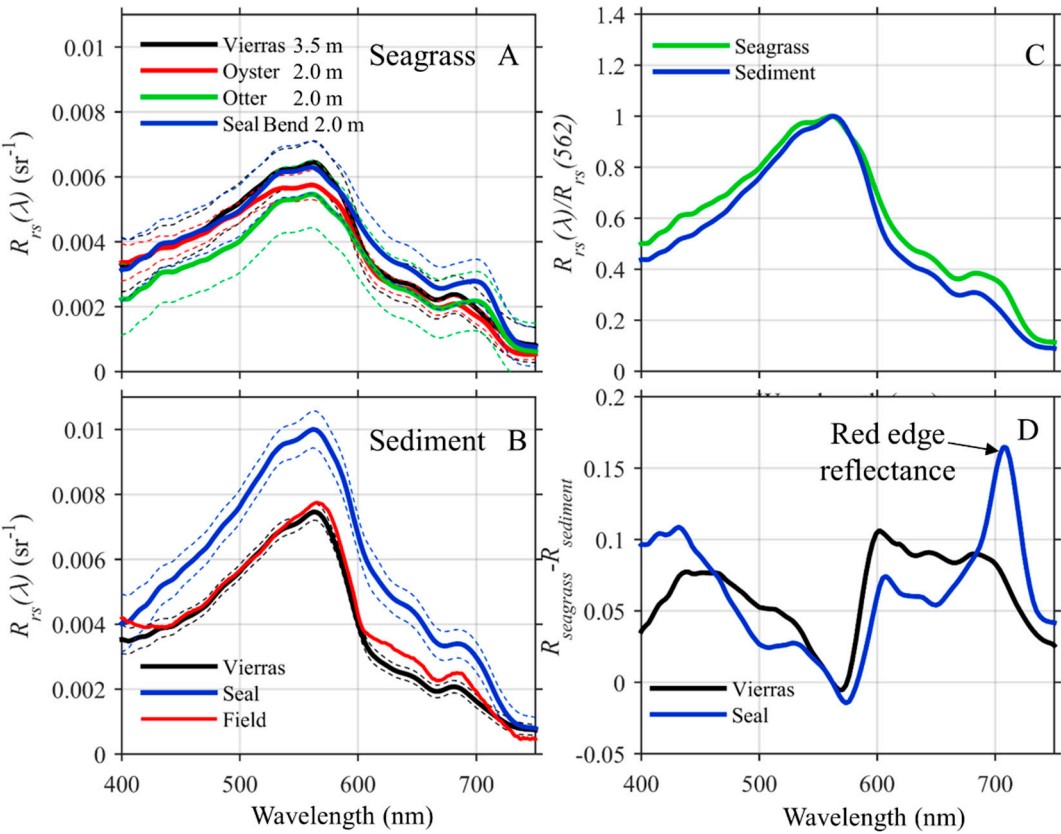

**Figure 9.** Mean spectral remote sensing reflectance (sr$^{-1}$) (solid lines) and one standard deviation from the mean (dashed lines) from the PRISM imagery measured overlying seafloor comprised of (**A**) eelgrass beds and (**B**) sediment. Approximate mean bed depths at high slack tide are provided. The field measurement of Rrs conducted over sediment is shown for comparison in panel B. (**C**) The mean PRISM spectra over eelgrass (from panel A) compared to over sediment (panel B) normalized at 550 nm to reveal spectral shape differences. (**D**) The difference between the mean eelgrass and sediment PRISM spectra at Seal Bend and Vierra eelgrass beds reveals similar patterns with the exception of the red edge of reflectance which is evident at the shallower Seal Bend site.

### 3.4.1. Red Edge and Threshold Algorithms

Because water molecules absorb highly in far red and near infrared wavelengths [66], the red edge reflectance of the seagrass can be absorbed by the overflying water column and become less visible at the sea surface with increasing water depth. This is particularly evident in Figure 9D, which shows the difference between $R_{rs}$ over seagrass and sediment from Vierra and Seal Bend. The red edge is dramatic for pixels overlying the shallow Seal Bend, whereas the red edge is barely evident at the deeper Vierra bed. This means that simple threshold algorithms to identify the presence of submerged aquatic vegetation like eelgrass using a decision tree approach [33,54] will only identify the shallowest seagrass beds, even though spectra may be similar across the blue and green portions of the visible spectrum (Figure 9A).

Various threshold techniques were applied to the mean spectra from the various eelgrass beds (Table 2). As shown graphically in Figure 9D, the standard red edge NDI using 700 nm and 670 nm was able to separate the shallower beds in Otter Curve and Seal Bend with a positive enhancement at 700 nm. However, the deeper Vierra and Oyster Alley beds did not have an enhancement at 700 nm, and the negative values would have been categorized as sediment. Using further near infrared bands (731 nm) in the ratio reduced performance, and no statistical difference was evident between sediment and seagrass for either shallow or deep beds. The $R_{rs}$ at 730 nm is related to the atmospheric correction methods for aerosols and sea surface reflectance and can be near 0 for many water types.

**Table 2.** Normalized difference indices (NDI) and ratios applied to remote sensing reflectance retrieved from the PRISM imagery over seagrass and sediment pixels.

| Region | Indices | | |
|---|---|---|---|
| | NDI(700, 670) | NDI(731.3, 673.6) | $R_{rs}(675)/R_{rs}(550)$ |
| *Seagrass Threshold* | *>0* | *>0.1* | *>0.35* |
| Vierra | ***−0.069*** [1] | 0.40 | 0.368 |
| Oyster Alley | ***−0.071*** | 0.53 | 0.357 |
| Otter Curve | 0.057 | 0.42 | 0.370 |
| Seal Bend | 0.041 | 0.43 | 0.418 |
| *Sediment Threshold* | *<0* | *<0.1* | *<0.35* |
| Vierra | −0.090 | ***0.40*** | 0.283 |
| Seal Bend | −0.059 | ***0.54*** | 0.342 |

[1] The values in bold italics are not within the threshold provided for this category and would be inaccurately classified.

Other simple reflectance ratios, like the red to green ratio shown in Table 2, can be useful for identifying different bottom types if the threshold is set precisely. However, determining a priori what threshold to use will depend on phytoplankton biomass in relationship to the benthic vegetation and other absorbing and scattering contributions from the water column.

### 3.4.2. Semi-Analytical Inversion Modeling

Semi-analytical inversion algorithms can be implemented to estimate bottom reflectance, as well as the bathymetry, and water column properties. Retrievals using a red edge algorithm [33] and results from the HOPE semi-analytical inversion model [10] were compared (Figure 10). This graphic demonstration of the results in Table 2 shows that the red edge algorithm was able to properly identify submerged aquatic vegetation in the shallow beds at Seal Bend and Oyster Curve but not the deeper Vierra bed or the eelgrass found in Oyster Alley.

The retrievals of the HOPE model were further evaluated over the Vierra bed where nearby coincident water column data was obtained (black box from Figure 10). The benthic reflectance of eelgrass and sediment for the Vierra bed (Figure 11A,B) were consistent with the overall input spectral shapes for these endmembers (see Figure 5). The eelgrass is darker and greener in color than the surrounding sediment, and the ratio between the chlorophyll-*a* absorption maxima around 675 nm and 550 nm provides a good outline the extent of the eelgrass bed (Figure 11C). The retrievals from HOPE were also similar to those modeled using a simple band ratio algorithm tuned to these waters as published in [23] (Figure 11D). Such a threshold approach is specific to the specific image and is not considered robust or portable to different water conditions and bathymetry.

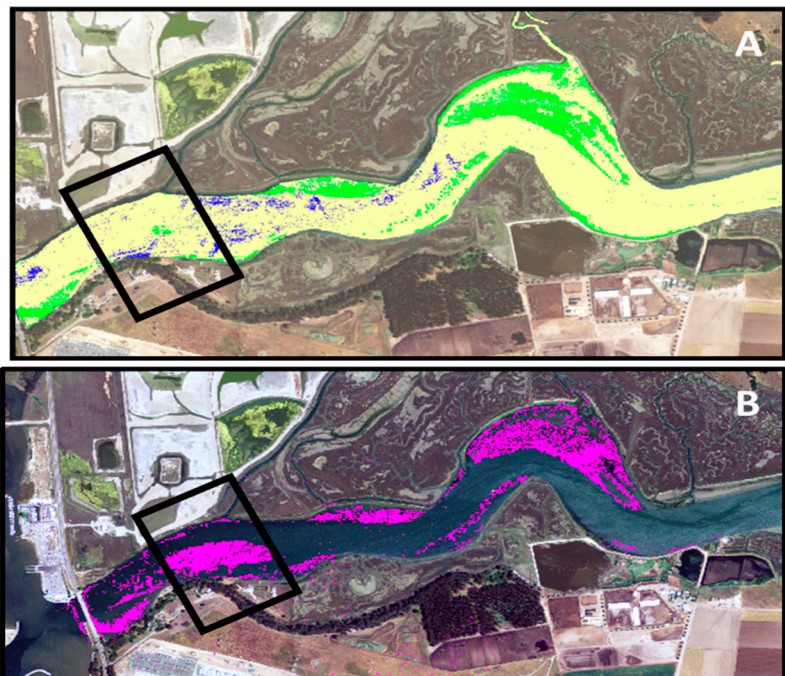

**Figure 10.** Habitat maps developed from the hyperspectral imagery using (**A**) a red edge threshold approach from Hill et al. (2017) and (**B**) the HOPE inversion model. The magenta pixels include those with benthic reflectance corresponding to eelgrass. The black box outlines the deeper Vierra bed which was only visible with the HOPE model.

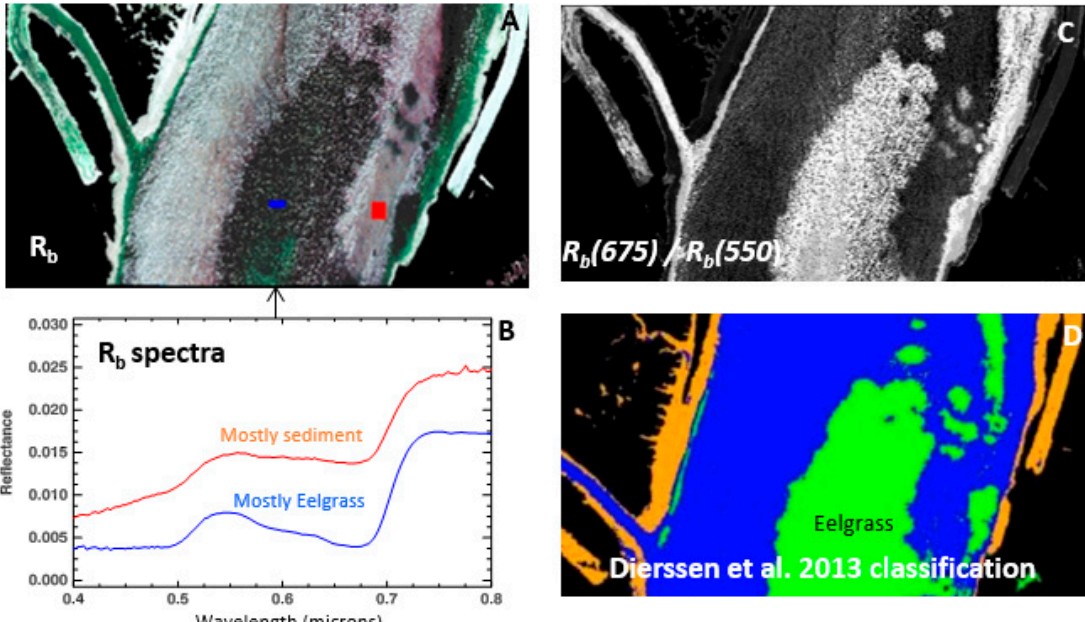

**Figure 11.** (**A**) Benthic reflectance retrieved over the Vierra bed retrievals shown in pseudo true-color (RGB) and (**B**) the corresponding spectral benthic reflectance across visible wavelengths for pixels identified in Panel A as dominated by eelgrass and sediment. (**C**) Relationship between benthic reflectance at 675 nm and 550 nm related to chlorophyll absorption is a good measure of the extent of the eelgrass bed. (**D**) A similar bed extent was mapped using a tuned chlorophyll-type algorithm with the same PRISM imagery [23].

Comparisons to measurements made over field transects averaged between 2011 and 2013 provide further validation of the accuracy of the HOPE model (Figure 12). The semi-analytical inversion

was able to effectively remove the influence of the water column overlying the bed (Figure 12A) and retrieved a detailed map of the underlying Vierra bed without using image-specific thresholds (Figure 12B). The eelgrass was clearly visible as having benthic reflectance that was darker and green-peaked compared to the surrounding seafloor. The green dots represented field measurements with high eelgrass percent cover (>20%); the light blue or cyan was (10% to 20%) and dark blue (<10%). In the Vierra bed, benthic vegetation was largely eelgrass, but nearshore stations can be dominated by the green macroalgae *Ulva* sp. or the fleshy red macroalgae *Gracilariaceae* and *Sarcodiotheca gaudichaudii*.

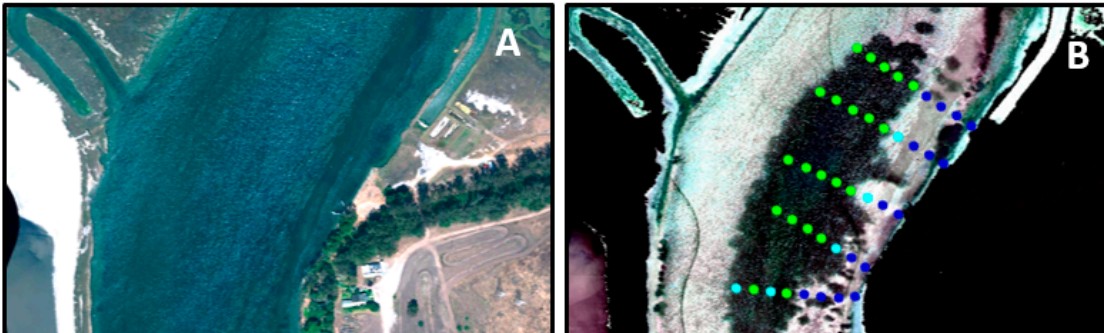

**Figure 12.** Pseudo true color images from the PRISM imagery over the Vierra eelgrass bed in Elkhorn Slough for (**A**) remote sensing reflectance Rrs and (**B**) retrieved benthic reflectance using the HOPE model. The dots represent a comparison of eelgrass density from diver field measurements showing stations with percent cover of Zostera marina <10 (blue), 10–20 (cyan) and >20 (green).

Retrieved water column properties of absorption by phytoplankton and dissolved and detrital material from the PRISM imagery were in close agreement with the field measurements and within the measured standard deviations (Table 3). Absorption of dissolved and detrital material $a_{dg}(440)$ dominated total absorption and averaged 0.31 m$^{-1}$ from the PRISM retrieval compared to 0.37 m$^{-1}$ measured in the field. The in situ $a_{ph}(440)$ and lab filtered chlorophyll-*a* values of 0.15 m$^{-1}$ and 4.0 mg m$^{-3}$ were also similar to the mean values retrieved from the PRISM data of 0.19 m$^{-1}$ and 4.75 mg·m$^{-3}$. The bathymetry retrieved from the PRISM imagery (2.6 ± 1.2 m) was also within the range published for the Vierra bed of around 2–4 m.

The major differences from the model were found in the retrievals of backscattering coefficient and then suspended particulate matter. The HOPE model retrieved a four-fold lower estimate of $b_{bp}(550)$ and TSM compared to field data (Table 3). Since the discrepancy is consistent between both measurements from backscattering sensor and bottle samples for TSM, it suggests that the error is not in the field measurements but in the model retrievals. However, if the backscattering was as high as the field measurements, the expected reflectance would also be much higher than those retrieved both in the field and from the PRISM imagery (see Figure 9). One possible explanation is that *Rrs* is indeed higher, and the atmospheric correction techniques for both the PRISM and above water measurements result in underestimates of Rrs. However, the 4-fold magnitude difference is likely too large to be explained by such correction techniques. Another possible explanation is that the water was indeed less turbid over the eelgrass bed than the field data collected downstream of the bed. Studies have found that eelgrass beds can reduce and divert current flow [67], and Buonassissi and Dierssen (2010) found particle fields to be lower over the dense seagrass beds compared to surrounding water [24].

**Table 3.** Comparison of HOPE retrievals of water column optical properties and bathymetry of the Vierra seagrass bed with concurrent field data.

| Parameter | PRISM Imagery | Field Data |
|---|---|---|
| $P$, $a_{ph}(440)$ (m$^{-1}$) | $0.15 \pm 0.04$ | $0.19 \pm 0.041$ |
| $G$, $a_{dg}(440)$ (m$^{-1}$) | $0.31 \pm 0.07$ | $0.37 \pm 0.09$ |
| $X$, $b_{bp}(550)$ (m$^{-1}$) | $0.016 \pm 0.005$ | $0.059 \pm 0.009$ [1] |
| $H$ (*Depth*) (m) | $2.58 \pm 1.21$ | $1.96$–$4.46$ [2] |
| Chlorophyll *a* (mg·m$^{-3}$) | $4.00 \pm 1.92$ | $4.75 \pm 0.72$ |
| TSM (g·m$^{-3}$) | $1.02 \pm 0.93$ [3] | $3.96 \pm 0.33$ |

[1] The field data averaged 0.046 m$^{-1}$ at $b_{bp}(660)$ and was adjusted to $b_{bp}(550)$ using a $Y$ of 0.5; [2] The depth range provided for the Vierra bed in Hammerstrom [40] at MLLW was reported as 0.5–3 m was adjusted with additional 1.46 m for high tide during the overflight; [3] Total Suspended Matter (TSM) is not a product of the HOPE model but was estimated from $b_{bp}(440)$ using an average $b_{bp}*$ of 0.0156 m$^2$·g$^{-1}$ [68], which is similar to the measured $b_{bp}*$ of 0.011 m$^2$·g$^{-1}$.

Finally, another contributing factor is that the benthic reflectance for eelgrass in the model was incorrectly represented as a horizontal eelgrass leaf. As highlighted in an optical closure study over a similar eelgrass bed in Long Island Sound [69], remote sensing reflectance is considerably lower than that estimated using the benthic reflectance of a flat seagrass leaf. From pictures taken in the field (Figure 13), the eelgrass canopies in Elkhorn Slough have shadows, aged leaves, and detrital material that result in a darker seafloor that absorbs more and scatters less light than a green eelgrass leaf. This explanation is also consistent with the higher sediment reflectance retrieved by the model (Figure 11B) than that measured in the field (Figure 5). Additional modeling with the HOPE algorithm on individual PRISM spectra confirmed that higher particulate backscattering in the water column would be retrieved with a lower benthic reflectance representative of an eelgrass canopy.

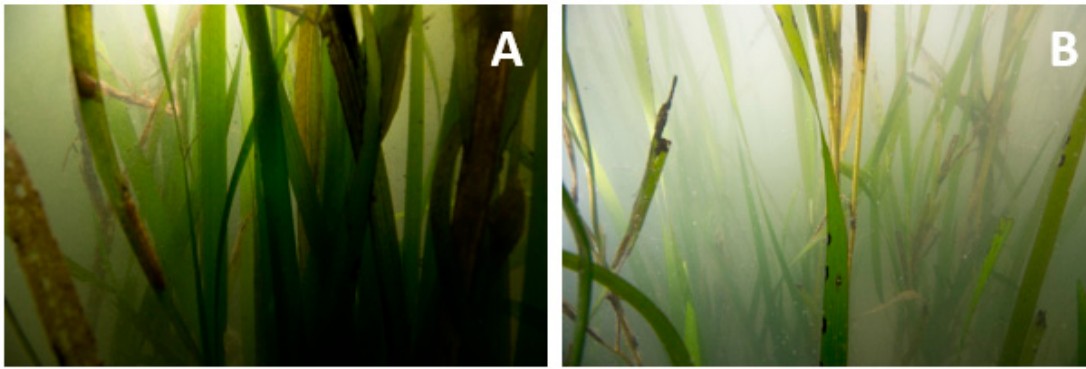

**Figure 13.** Photographs taken of eelgrass in Elkhorn Slough showing the eelgrass canopy in the turbid water. This illustrates how a single benthic reflectance corresponding to a green leaf is not a good proxy for the three-dimensional architecture and variability of shoots within the canopy. Photo credit Eric Heupel.

## 4. Conclusions and Outlook

Remote sensing of submerged vegetation in coastal waters presents unique challenges relative to terrestrial systems. Photons predominantly penetrate downward into the water column, and the fraction of photons that leave the water column is typically only a small fraction of the signal measured at a satellite or aircraft. Hence, water targets are generally much darker than adjacent land surface. This is particularly true when light-absorbing benthic vegetation is present. Analysis of Google Earth imagery revealed that submerged vegetation appear as dark patches compared to surrounding water. However, the ability to detect the optical signal from a submerged eelgrass canopy is related to the water depth and clarity as well as the amount of radiance (related to time of day, latitude, and season) and the relative amount of glint or sea surface reflection in relation to the water-leaving signal.

Hence, the signal from the water column and benthos can be overwhelmed by the contributions reflected from the sea surface and atmosphere. Such was the case for the Google Earth analysis conducted here. Due to differences in image quality, photointerpretation of RGB imagery from the historic archive could not be effectively used to accurately delineate the different eelgrass beds in Elkhorn Slough over time. For the time span from 2011–2013 when field samples showed no statistical difference in the Seal Bend and Vierra bed, the imagery was highly variable showing beds disappearing and reappearing over time. It was not a single bad image but rather inconsistencies between all of the imagery that made it impossible to construct an accurate time series.

The calibrated and atmospherically-corrected PRISM imagery allows for robust approaches for mapping benthic vegetation. In the absence of an overlying water column, the normalized difference vegetation index (NDVI), a ratio of the near infrared and visible bands, can be effectively used to detect green biomass [70]. In oceanic waters, however, the infrared portion of the spectrum is highly absorbed by the water column, and red edge algorithms for detecting submerged vegetation are limited to shallowest beds. The results here confirm that such threshold approaches were only able to discern the shallowest eelgrass beds in Elkhorn Slough. Unlike regions with clear water and bright carbonate sand (e.g., Bahamas Banks [64]), this coastal habitat with dark sediment and turbid water proved challenging for discerning the Vierra bed in the deeper part of the main channel.

The semi-analytical HOPE model, however, was able to accurately delineate the extent of all of the five identified beds within this portion of Elkhorn Slough, including the deeper Vierra bed. The retrieval of eelgrass in the Vierra bed was validated from transects measured in the field, but the match between 1-m pixels and underwater locations continues to be a challenge for such analyses. Given the way light refracts underwater, as well as the lack of GPS underwater and accuracy of geolocation in imagery, new techniques will need to be developed to accurately validate 1-m pixel imagery. This conclusion was also reached by Hedley et al. 2017 in an analysis of 1-m PRISM imagery from Florida Bay. A sensitivity analysis revealed that a primary limitation to the study was spatial alignment with the imagery with in situ data, which was heterogeneous on scales of a few meters [11]. In that case, the ability to retrieve the appropriate range in seagrass density or LAI for a specific location was used rather than a single point measurement. Validating satellite measurements with even higher spatial-resolution imagery from drones and techniques like fluid lensing, for example, may also prove useful [71].

Both the water column absorption properties and bathymetry were shown to be retrieved quite accurately with the HOPE model. Providing further confirmation of the modeling results [11], retrievals of bathymetry are quite robust to noise and assumptions about water column and benthic composition. The model produced a nearly accurate estimate of water column chlorophyll-*a* and absorption by colored dissolved and detrital matter. Retrieved particulate backscattering was much lower than that measured in the field. A contributing factor was likely the representation of benthic reflectance as a flat green eelgrass leaf rather than a 3-D, shadowed, detritus-covered canopy. As shown in Hedley et al. (2018), for example, the benthic reflectance of a 3-D coral is generally much lower than that measured from a horizontal plane and can be related to an estimate of the true surface area divided by the planar (e.g., rugosity). However, measurement and representation of more realistic benthic reflectance that incorporates the rugosity [72,73] or canopy architecture [74] and possibly epiphytes and detritus is warranted to reduce the uncertainty in retrievals of all water column parameters.

On a positive note, however, modeling benthic type and bathymetry are fairly robust to such scattering errors. Since the spectral shape of backscattering is fairly flat, incorporating high scattering in the seafloor rather than water column does not significantly influence the retrieval of the spectrally varying constituents such as bottom type and absorption by phytoplankton nor influences retrievals of bathymetry. The HOPE model used here was parameterized for seagrass and sediment, but other benthic constituents could be added to provide additional information on other submerged aquatic vegetation such as the green and red macroalgae that occurred along the shore and mixtures of bottom types [59]. Various different permutations of semi-analytical inversion models have been

evaluated with different numbers of bottom types, and analytical solutions have been developed in recent years [12,59,75]. Comparing such approaches widely across different ecosystems would be an important next step.

The high-resolution maps produced here may be considered as a baseline for future studies monitoring changes in the Elkhorn Slough environment and assessing water quality variability due to nutrients and natural and man-made changes to flow within this estuary. Additionally, the long-term impact of the presence of top predators on the eelgrass distributions can also be assessed [18]. This study illustrates how hyperspectral radiometry can be used to provide accurate monitoring of both the water quality and benthic habitats, even in eutrophic estuarine waters.

**Author Contributions:** Conceptualization, H.M.D.; methodology, H.M.D., D.R.T., Z.L.; software, D.R.T., Z.L.; validation, H.M.D., K.H., K.J.B., A.C.; formal analysis, A.C., D.R.T., Z.L.; resources, H.M.D.; writing—original draft preparation, H.M.D., D.R.T., K.J.B.; writing—review and editing, H.M.D., K.H., D.R.T., Z.L.; visualization, A.C., D.R.T., K.H.; project administration, H.M.D.; funding acquisition, H.M.D.

**Funding:** This research was funded by the National Aeronautics and Space Administration through the Coral Airborne Remote Sensing Laboratory (CORAL) (NASA Grant number NNX16AB05G), NASA's Ocean Biology and Biogeochemistry Program, Earth Science and Technology Office, and the Airborne Science program for development of the PRISM sensor.

**Acknowledgments:** We would like to thank the individuals who contributed to collection of field data including Jeff Godfrey, Eric Heupel, and the small boat operations at Moss Landing Marine Laboratories. We also thank the NASA Jet Propulsion Laboratory PRISM team including Pantazis Mouroulis, Ian McCubbin, and the pilots for airborne imagery collection. Alexandre Castagna is acknowledged for the LISST processing. Bathymetry data used in this study were acquired by the Seafloor Mapping Lab of California State University Monterey Bay. A portion of this research was performed at the Jet Propulsion Laboratory, California Institute of Technology. US Government Support Acknowledged.

**Conflicts of Interest:** The authors declare no conflicts of interest.

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
