# Peer review of "Pushing the Limits of Seagrass Remote Sensing in the Turbid Waters of Elkhorn Slough, California"

_remotesensing, doi:10.3390/rs11141664_

Round 1

Reviewer 1 Report

The paper is a solid well written scientific piece of work, with a strong treatment of aquatic optics and shallow water remote sensing along with thorough and appropriate references.

The specific focus on shallow habitat mapping in turbid waters does separate it from the majority of studies in this field, and makes it a worthy contribution to the literature.

I do have a couple of broader suggestions around scope of the work that I think would improve and  reinforce the aims and conclusions the authors are able to make. These are discussed below, and followed with a few more specific clarifications I think are required around the methodology.

Broad Issues & Suggestions

Selection of data sources and methods

With the stated aim of 'evaluating the advantages and limitations of different remote sensing approaches', the jump from RGB image interpretation through to physics-based inversion of airborne hyperspectral data seems to be quite a large one, leaving out many options in between. The use of multi-spectral satellite data seems to be dismissed very early, largely on the basis of the spectral resolution of two older sensors, and the fact they are unable to discriminate between seagrass, macroalgae and coral (lines 37-42). 

Given that the study focuses specifically on distinguishing only between seagrass and sediment, it would seem this rationale is not really appropriate. A newer sensor like Sentinel-2 would appear like an appropriate option to be evaluated, especially given the inclusion of the red-edge bands on S2 which are discussed in the manuscript as very relevant in picking out seagrass from sediment.

Leaving out the evaluation of an EO option such as this seems a bit of a gap in my opinion, particularly in the context of the aim of ongoing monitoring and the massive cost difference between an airborne campaign and free EO data. 

Discussion of LAI

A decent part of the manuscript is spent discussing LAI, before the (well-justified) decision to not attempt to quantify it from the imagery analysis. It leaves me unsure if the level of discussion of LAI in the context of this study is then needed, and it may add extra unrequired content/length to the manuscript.

RGB image visual interpretation

This method as a whole seems very subjective, and for example would be very dependent on how the imagery was histogram stretched.

I do understand the point the authors are trying to make in terms of variable image quality to reinforce the potential of hyperspectral corrected data. However I do think that point would be stronger if, as mentioned above,  the benefits were framed over other actual EO spectral data (e.g Sentinel 2, World View 2 etc).

Method details and clarifications

Lines 233 to 237 – the at-sensor reflectance notation seems to be inconsistent (different symbology for rho?)

Lines 281 to 289 – Perhaps a short line describing how the U matrix and v vector relate to the earlier described parametrisation of B and then bottom reflectance in the HOPE parameters.

Lines 296 to 299 – It is understandable to try and regularise the inversion, but I think more detail is required to explain how the priors of the expected value u and standard deviation for each parameter are derived. In lines 156-157 it is stated that the field work values are not used for calibration of the algorithms. Given that HOPE is an independent pixel by pixel inversion, how then are sensible expected values for each pixel e.g. water column depth and optical properties, assigned prior for use in the SSE?

Line 297 – How is the measurement noise derived? Is this an image based ‘environmental’ noise such as NeDR? Does it incorporate the covariance noise of the bands?

Line 440-441 – This seems to referring to rrs rather than Rrs? Maybe a clarification and line about why Rrs was used for most analysis, but rrs for input into HOPE.

Lines 496 – When mapping the eelgrass extent from HOPE, how was the v vector solution used to determine whether a pixel was grass or not? E.g was it when more than 50% of the mixing fractions came from a combination of the three eelgrass spectra used in the library? Was the derived bottom reflectance spectra matched back to a single seagrass reference spectra?

Reviewer 2 Report

Great majority of benthic habitat mapping studies have been carried out in optically clear ocean waters. The full potential of remote sensing is still to be exploited, particularly in optically complex waters with high turbidity or CDOM concentration. As there are large amount of researchers working in complex water environments, the current paper would be of great interest for them.

Some minor remarks and clarifications needed:

Page 4, line 128: “Additionally, the field validation effort attempted…“ Why do the authors refer to overpasses of ocean colour sensors (VIIRS NPP, MODIS) if satellite data was not used in the current study?

Page 4, line 130:  “The closest field measurement...“  Not clear if authors have meant the overpass of PRISM or overpass of MODIS, VIIRS NPP (the latter were mentioned in the previous sentence).

Page 4, line 130: Please provide additional information about field stations marked as PR... Currently it’s not clear whether those stations refer to temporal stations or spatial stations? What was the time interval for measurements?

Page 4 and 5, Figure 2 and 3. The eelgrass bed in Figure 2 is named Seal Alley, while the same bed in Figure 3 is named Oyster Alley.

Page 5, line 168: “Samples were collected at 1 m depth…“ Not clear if data were collected only at the single depth of 1 m? Check Figure 7, which shows profiling data from the LISST from 0.0-10 m.

It is a bit confusing to understand, which measurements were performed from boats (spatial stations) and which ones were performed by buoy (temporal stations)? Were there any water column measurements from boats?

Page 9, line 308: “The final analysis is conducted over the Vierra eelgrass bed…“ Once again confusing sentence. In page 4, line 130 you state that the closest field measurements to the overflight were conducted in the location of the West LOBO buoy, not over the Vierra eelgrass bed. If there were different boats at different locations at the time of PRISM overflight, then it should be specified in the methods chapter.

Page 10, Figure 5. It is said that the reflectance of eelgrass leaf is measured in the laboratory, but in Methods chapter only underwater reflectance measurements were described.  

Page 11, line 368: “Overall, absorption was low compared to total attenuation…“ I would like to see a graph showing the spectral absorption compared to attenuation.

Page 14, Figure 9A and 9B. Please include average water depths of each seagrass bed to the graphs, so that it would be easier to follow the difference in spectral values.

You refer to shallower and deeper seagrass beds. Please provide average estimated depths for each bed.

Page 14. Figure 9. Would it be possible to provide some more examples comparing PRISM and in situ measured spectra to assure that atmospheric correction is high-quality?

Page 18, line 554: “Finally, another contributing factor...”. In Methods chapter you described that seagrass canopy underwater measurements were completed in the study area. Why weren’t those spectra used as endmembers in the HOPE model?
